# Effect of Post-Weld Annealing on Microstructure and Growth Behavior of Copper/Aluminum Friction Stir Welded Joint

**DOI:** 10.3390/ma13204591

**Published:** 2020-10-15

**Authors:** Yuhua Jin, Bo Wu, Xuetian Lu, Yichu Xing, Zizheng Zhou

**Affiliations:** 1State Key Laboratory of Advanced Processing and Recycling of Nonferrous Metals, School of Materials Scienceand Engineering, Lanzhou University of Technology, Lanzhou 730050, China; bowu0118@163.com (B.W.); 18332179137@163.com (X.L.); yichuxing668@163.com (Y.X.); zizhengzhou1@163.com (Z.Z.); 2School of Materials Science and Engineering, Lanzhou University of Technology, Lanzhou 730050, China

**Keywords:** aluminum alloy, copper, friction stir welding, annealing, intermetallic compounds

## Abstract

Friction stir welding of 1016 pure aluminum and T2 pure copper with 2 mm thickness was carried out in the form of lap welding of copper on the upper side and aluminum on the lower side. The growth of interface microstructure between 1016 pure aluminum and T2 pure copper welded by friction stir welding was studied. The growth mechanism of the intermetallic compound (IMC) layer in the Cu-Al lap joint was revealed by annealing at 300, 350, 400 °C. The intermetallic compound (IMC) layer in the lap joint grows again during annealing, and only the original structure of the intermetallic compound (IMC) layer grows at lower annealing temperature and holding time. At higher annealing temperature and holding time, the original structure of intermetallic compound (IMC) layer no longer grows, and a new layered structure appears in the middle of the original structure. There is a gradient change of microhardness in the nugget zone. With different holding times, different softening phenomena appear in the metals on both sides of copper and aluminum. When the hardness decreases to a certain extent, it will not continue to decrease with the increase of holding time. When the annealing temperature is 350 °C and 400 °C, the strength of the tensile sample increases first and then decreases with the increase of holding time. At the interface of Cu-Al, the fracture runs through the whole intermetallic compound (IMC) layer.

## 1. Introduction

Aluminum and copper are widely used in aerospace, electronics, chemical and other fields because of their excellent electrical and thermal conductivity [1,2,3,4]. In the process of equipment processing and manufacturing, the problem of connecting the dissimilar metals aluminum and copper often arises [5,6,7,8]. At present, the traditional fusion welding method is usually used for the dissimilar materials of aluminum and copper. Because there are obvious differences in the physical and chemical properties between the two materials, in the welding process, defects such as cracks, pores, and coarse tissues are prone to occur, resulting in a decrease in the strength of the joints, and even a loss of effectiveness in the connection [9]. Friction stir welding is an environmentally friendly welding technology with high welding quality and low production cost. It has been rapidly applied in the welding of aluminum alloy, magnesium alloy, copper alloy and other materials. The connection is mainly realized by the plastic flow of high-temperature metal, which effectively avoids the solidification of structural defects such as pores and inclusions caused by melting welding. It has outstanding technical advantages in dissimilar metal connection [10,11,12,13,14,15]. However, connections between aluminum–magnesium, aluminum–copper and other dissimilar materials are still in the exploratory research stage. Therefore, some people are beginning to explore this method for the purpose of welding dissimilar materials such as aluminum–copper, with a focus on the microstructure and properties of the joint. At present, the material system of dissimilar metal friction stir welding (FSW) for aluminum–copper carried out by scholars at home and abroad is focused on red copper and 1, 5 and 7 series aluminum alloys [16,17,18,19,20,21]. Although friction stir welding can avoid the problems of high temperature oxidation, element burning loss, cracks, and pores in Cu-Al welding, it is still easy for intermetallic compounds to appear. Controlling the generation of intermetallic compounds and strengthening the properties of welded joints through process improvement and interface research is the research focus of dissimilar metal friction stir welding for Cu-Al at the present stage. Raju Prasad Mahto et al. [22] studied the effects of inclination and rotational speed on welding defects, intermetallic compounds, welding strength and microhardness in the friction stir welding of AA6061-T6 and AISI304 sheets. The thickness of the intermetallic compound (FexAly) layer in the weld zone varies unevenly with the process parameters. With increasing porosity of the weld, the defects of the steel hook are enlarged. With increasing thickness of the intermetallic layer, there is a significant decrease in joint strength and elongation. Avettand-Fènoël et al. [23] investigated the microstructure and mechanical properties of linear friction welding of AA2024–pure copper. In this paper, the microstructure characteristics of close contact with the interface are studied in depth, and a phenomenological mechanism of material flow is proposed. The interface is covered with a layer of discontinuous intermetallic compounds. The interface is covered with a layer of discontinuous intermetallic compounds. Al_2_Cu, AlCu, Al_2_Cu_3_, and especially Al_4_Cu_9_ were detected. Two metastable phases—Al_3_Cu_2_ compound and non-equilibrium Al solid solution containing 13 at.% Cu—were found at the interface and on the Al side, respectively. According to the strain state in the welding process, an ideal shear texture is formed on both sides of the joint. Due to the formation of a large number of intermetallic compounds, the mechanical resistance of the joint needs to be improved. Sachindra Shankar et al. [24] studied the successful preparation of AA1050 and oxygen-free copper with different thicknesses by friction stir welding. In the interface structure of Al-Cu, a layer of Al_4_Cu_9_ and Al_2_Cu intermetallic compounds (IMCs) with a thickness of 2.2–0.26 μm was formed at different welding speeds. Intermetallic compounds (IMCs) with large volumes had an adverse effect on the tensile properties of the joint. When the welding speed was 160 mm/min and 213 mm/min, and the tool offset was 1 mm, the intermetallic compounds (IMCs) (0.26–0.77 μm) had little effect on the tensile properties. Lee et al. [25] annealed the steel, aluminum and copper composite plate prepared by hot rolling process, and analyzed the interface structure and mechanical properties. The tensile strength of the composite plate decreased linearly, the elongation increased slightly with the increase of annealing temperature, and the elongation decreased at 400 °C as a result of the brittle intermetallic compound layer. Won-BaoLee et al. [26] realized welding between pure copper and 1050 aluminum through traditional friction welding, and further analyzed the growth law of copper–aluminum intermetallic compounds in the joint and its effect on the mechanical properties of the joint after annealing. It was found that the tensile strength decreased with the increase of intermetallics, and the fracture position changed from the heat-affected zone on the side of Al to the whole intermetallic layer. Although some progress has been made in the study of the effect of annealing temperature on the interface of Cu-Al, the related research is still not systematic. 1060 pure aluminum and T2 copper with a thickness of 2 mm were treated by friction stir welding (FSW), and the microstructural characteristics of the joints were characterized. This paper focuses on the analysis of the relationship between microstructure and mechanical properties of Cu-Al lap joints before and after annealing, in order to provide theoretical guidance for obtaining high-quality Al-Cu dissimilar metal friction stir welding (FSW) connections.

## 2. Materials and Methods

The experimental material used 1016 pure aluminum (Zhejiang Zhongyu Electric Co., Ltd., Wenzhou, Zhejiang, China) and T2 pure copper (Zhejiang Zhongyu Electric Co., Ltd., Wenzhou, Zhejiang, China) in rolling state. The size of the plate was 200 mm × 100 mm × 2 mm. Its chemical composition is shown in Table 1 and Table 2. The oil stain on the workpiece surface was wiped off with acetone before welding. The lap welding of the copper plate on the upper side and the aluminum plate on the lower side was carried out as shown in Figure 1a. The welding equipment comprised a friction stir welding machine of the model FSW-3LM-015 (Sooncable, Beijing, China). As shown in Figure 1b, the mixing head was made of H13 tool steel. The length of the mixing needle was 2 mm. The diameter of the root of the mixing needle was 4.7 mm. The diameter of the end was 3.5 mm. The diameter of the shoulder was 15 mm. The shoulder was concave, and the inner concave angle was 2°. The post-weld joint was obtained by carrying out the welding process at a rotation rate of 1100 r/min at a welding rate of 50 mm/min. The friction stir welding (FSW) joints were treated by annealing, and the effects of different annealing temperatures and holding times on the growth behavior of the intermetallic compound (IMC) layers were studied. The annealing temperatures used were 300 °C, 350 °C and 400 °C, and the annealing times were 0.5, 1.5, 3.5 and 4.5 h. The hardness test was carried out on a microscopic Vickers hardness tester (Wenzhou Fangyuan instrument Co., Ltd., Wenzhou, Zhejiang, China). The tensile specimen was prepared by cutting it perpendicular to the weld. The tensile test was carried out using a universal electronic testing machine, and the tensile fracture morphology was observed using a Quant450 field emission scanning electron microscope (FEI, Hillsboro, OR, USA).

## 3. Results

### 3.1. Macroscopic Morphology of the Cu-Al Interface

Figure 2a shows the surface morphology of representative dissimilar Al-Cu joints manufactured at a rotational speed of 1000 r/min and a transverse speed of 50 mm/min.

In the figure, it can be observed that there are no macroscopic defects such as tunnels inside the weld, and a hook structure is formed at the edge of the stirring zone at the interface between the aluminum and copper. This structure is one of the typical characteristic structures of friction stir welding lap joints of dissimilar metals [27]. When analyzing the characteristics of the hook structure, it was found that the structure extends from the copper side to the aluminum side. According to the performance characteristics of aluminum and copper, it is known that the melting point of copper is high, and the fluidity is poor in the welding process. In the process of the action of the stirring head on the copper plate in the thermomechanical influence zone, part of the copper will tear from the matrix (although not completely detach) and extend into the aluminum side under the stirring action of the stirring head, finally forming a hook structure. When the copper plate is placed at the top, the hook structures of both the forward side and the back side bend to the inside of the weld, and the main flowing metal is copper. The copper in the affected zone of the Cu-Al interface is reduced by the extruding pressure to the outside. At this point, under the action of the rotation of the stirring head, it bends to the inside of the weld, finally forming the structural characteristics of the inside of the weld, as shown in Figure 2b. To further analyze the influence of the position of the plate on the microstructure of the lap joint, the copper–aluminum joining interface is selected as the key analysis area to further analyze the specific structure of the lap joint.

### 3.2. Interface Micro-Morphology of Cu-Al

#### 3.2.1. Microstructure of Cu-Al Interface before Heat Treatment

A large number of crystalline structures can be observed under the copper–aluminum interface in Figure 3a, formed by the instantaneous liquefaction and re-solidification of the metal. Placing the copper plate on the top of the lap joint will increase the peak temperature of the weld, and when the peak temperature is higher than the melting point of aluminum, it will melt instantly, and the metal in the melting zone will cool and crystallize again with the advance of the stirring head. This eventually leads to the emergence of a eutectic structure. As shown in Figure 3b, combined with the interface position of Cu-Al, a single-layer strip structure is found, and the adjacent layer structure is irregular, extending downward in a similar fashion to the columnar crystal structure. As shown in Figure 3e, the Energy Dispersive Spectroscopy (EDS) test results show that the proportions of copper and aluminum in the layered structure are 61.9% and 38.1%, respectively, and the ribbon structure is composed of CuAl + CuAl_2_. As shown in Figure 3d, the proportions of copper and aluminum in the lower structure, similar to the columnar crystal, are 33.6% and 66.4%, respectively. The structure is composed of an aluminum-based solid solution of CuAl_2_ and copper. In Figure 3c, a granular structure can be observed in the crystallization region, and the composition of this structure is the same as that of the second layer, which is the result of the diffusion of aluminum atoms to copper particles. Under the action of the stirring head, the copper particles separated from the upper copper plate will enter into the lower aluminum plate, but when the copper particles appear in the liquid phase region, the freely moving aluminum atoms will enter the copper particles. When the aluminum atoms enter the copper particles completely, the original copper particles will be transformed into intermetallic compound (IMC) particles.

#### 3.2.2. Microstructure of Cu-Al Interface after Heat Treatment

Figure 4 shows the micro-morphology of the intermetallic compound (IMC) layers obtained at a fixed annealing temperature of 300 °C and holding times of 0.5 h, 1.5 h, 3.5 h and 4.5 h. During the analysis of the process characteristics of Cu-Al dissimilar metal friction stir welding, it was observed that an intermetallic compound (IMC) layer will be formed at the joining interface of Cu-Al. The intermetallic compound (IMC) layer has a two-layer structure, and there are some differences in the composition of intermetallic compounds (IMCs) in each layer. The microstructure of the intermetallic compound (IMC) layer can be observed at different holding times in Figure 4. When the holding time is 0.5 h, the intermetallic compound (IMC) layer is still a two-layer structure, which is basically the same as the unannealed joint, as shown in Figure 4a. When the holding time is increased to 1.5 h, there are two layers at the interface, but the thickness of each layer increases, as shown in Figure 4b. When the holding time is 3.5 h, the original layer 1 and layer 2 structures still exist, and the third layer structure appears between layer 1 and layer 2, which is marked as layer 3 in the figure, as shown in Figure 4c. When the holding time is 4.5 h, the intermetallic compound (IMC) layer has a three-layer structure, and the thickness of the three layers increases as shown in Figure 4d. At a fixed annealing temperature, the original structure thickness of the original intermetallic compound (IMC) layer increases with the increase of holding time, and a new layered structure appears in the middle of the original structure.

Figure 5 shows the micro-morphology of the intermetallic compound (IMC) layer obtained at a fixed annealing temperature of 350 °C with holding times of 0.5 h, 1.5 h, 3.5 h and 4.5 h. It can be observed that the structure of the intermetallic compound (IMC) layer is a two-layer structure when the holding time is 0.5 h, while the structure of the intermetallic compound (IMC) layer is a three-layer structure when the holding time is 1.5 h, 3.5 h and 4.5 h, whereby the third layer appears in the middle of intermetallic compound (IMC) layer, and the growth process of the intermetallic compound (IMC) layer is similar to that with an annealing temperature of 300 °C. With a short holding time, there is only growth of the original intermetallic compound (IMC) layer structure, and no new structure appears, because at a fixed annealing temperature and a short holding time, the diffusion number and range of copper and aluminum atoms are limited to a certain extent, and the conditions for the formation of a new layered structure cannot be achieved. With increasing holding time, the diffusion quantity and diffusion range of copper and aluminum atoms also increase, and the middle area of the original structure meets the material and thermodynamic conditions for the formation of a new layered structure, thus promoting the formation of a new layered structure. Comparing the newly formed intermetallic compound (IMC) layered structure at 1.5 h, 3.5 h and 4.5 h, it can be seen that among the three layered structures, the growth rate of the second layer is the fastest, and the growth rates of layer 1 and layer 3 are relatively slow. By comparing the positions of the three layered structures, it can be found that the second layer is located at the lowest end of the intermetallic compound (IMC) layer and is adjacent to one side of the aluminum plate. According to the physical properties of copper and aluminum, the melting point of copper (about 1084 °C) is much higher than that of aluminum, so if the melting point of aluminum has not been reached, the atomic activity of aluminum will be much greater than that of copper. The second layer is near the side of the aluminum plate, which means that the active atoms in the second layer are larger than those in the first and third layers, and the number of active atoms is conducive to the growth of the layered structure. This is why the growth rate of layer 2 is faster than those of layer 1 and layer 3.

Figure 6 shows the micro-morphology of the intermetallic compound (IMC) layer obtained at a fixed annealing temperature of 400 °C and holding times of 0.5 h, 1.5 h, 3.5 h and 4.5 h. It can be observed that the interface structure of the intermetallic compound (IMC) layer is a three-layer structure when the annealing temperature is 400 °C, and the thickness of the three layers increases with increasing holding time. By comparing the microstructure of the intermetallic compound (IMC) layer annealed at 300 °C, it can be seen that there is a certain relationship between the growth behavior of intermetallic compound (IMC) layer and the annealing temperature and holding time. Increasing annealing temperature or time can promote the growth of the intermetallic compound (IMC) layer. The growth of the intermetallic compound (IMC) layer includes two aspects: on the one hand, the thickness of the original layered structure increases with increasing annealing temperature or holding time; on the other hand, when the growth environment meets certain requirements, the layered structure of the original intermetallic compound (IMC) layer changes, and a new layered structure is formed in the middle of the original layered structure.

The annealing holding time is fixed at 4.5 h, and lap joints with annealing temperatures of 300 °C, 350 °C and 400 °C are selected to observe the morphology of the microstructure in the “liquid phase zone” of the joints. The evolution process of the structure in this area during annealing is studied in comparison with the joints without annealing heat treatment. By comparing and observing the structural characteristics of crystals in different “liquid regions”, it is found that the structural changes in this region are mainly manifested in two aspects: on the one hand, the structure adjacent to the “liquid region” in the intermetallic compound (IMC) layer at the Cu-Al joining interface grows to the “liquid region”, and evolves from the original columnar crystal structure into a layered structure. On the other hand, there are dendrites in the “liquid phase zone” structure of the unannealed joint. With the progress of annealing, the original dendritic structure gradually disappears, leaving a dense particle structure. Comparing the growth environment of the tissue in the welding process with that in the annealing process, it can be seen that there is an obvious difference between them. Excessive heat input during welding leads to the appearance of a “liquid zone” inside the joint, and the growth direction and growth mode of the microstructure in liquid environments have a certain selectivity. It can also be seen in Figure 7a that the growth of tissue has a certain directionality. During annealing, the growth of the microstructure is in a solid environment. Under the thermodynamic conditions of tissue growth, crystal grown in this way only depends on mutual diffusion between atoms. Crystal grown in this way will make up for the difference in the previous structure, making each region of the intermetallic compound (IMCs) interface grown in the “liquid zone” more uniform, changing the difference in the internal composition of the dendrite, and forming a dense granular structure as shown in Figure 7c.

Figure 8 shows the X-Ray Diffraction (XRD) test results of the joint at a fixed annealing temperature of 1.5 h and annealing temperatures of 300 °C, 350 °C and 400 °C. The results show that in addition to conventional copper and aluminum, oxides are also found. Oxides are produced in the joint during welding. In addition, there are many kinds of intermetallic compounds (IMCs) in the joint, and there are some differences in the intermetallic compounds (IMCs) generated in the joint under different annealing conditions. When the annealing temperature is 300 °C, the resulting intermetallic compounds (IMCs) are Al_2_Cu, AlCu and Al_4_Cu_9_.

The microstructure of the intermetallic compound (IMC) layer in the joint was observed with a holding time of 1.5 h and an annealing temperature of 300 °C, 350 °C, and 400 °C. The EDS measurements of different structures in the intermetallic compound (IMC) layer show that, as shown in Figure 9, the proportion of copper decreases and the proportion of aluminum increases gradually from top to bottom. Combined with the binary phase diagram of Cu-Al, it is known that the composition of the uppermost intermetallic compounds (IMCs) adjacent to the copper side and the intermetallic compounds (IMCs) adjacent to the aluminum side group into Al_4_Cu_9_ + AlCu, becoming Al_2_Cu and an aluminum-based copper solid solution, and the intermetallic compounds (IMCs) formed in the intermediate structure of intermetallic compounds (IMCs) group into AlCu + Al_2_Cu, and are affected by the proportion of copper and aluminum in this range. From the above observation results, it can be seen that different intervals of intermetallic compounds (IMCs) are composed of different kinds of intermetallic compounds (IMCs), and the types of intermetallic compounds (IMCs) found in the corresponding interval are affected by the proportion of copper and aluminum elements in this interval. In the process of Cu-Al dissimilar metal friction stir welding, copper and aluminum combine under the action of friction heat and the stirring head, and intermetallic compounds (IMCs) are formed at the joining interface due to the mutual diffusion taking place between copper and aluminum atoms. In the process of diffusion, copper atoms enter into the aluminum matrix, and the corresponding aluminum atoms also enter into the copper matrix. Due to the limited diffusion of elements, the proportion of dissimilar metal elements diffusing into each other is relatively low. In this case, two structural forms will be formed, that is, the two-layer metal intermetallic compound (IMC) layer observed in this paper. The effect of the temperature field during annealing will once again promote the diffusion of copper and aluminum atoms, and with increasing number and depth of copper and aluminum atoms, the intermetallic compound (IMC) layer will grow and the thickness of the original structure will increase. With increasing annealing temperature and holding time, the atomic activity will increase, and the difference in the atomic ratio of copper to aluminum in the central region will decrease during diffusion. According to the binary phase diagram of Cu-Al, the type of intermetallic compounds (IMCs) formed under the ratio of copper to aluminum is different from the original intermetallic compound (IMCs) structure, so a new layered structure is formed in this region, which changes the original structure of the intermetallic compound (IMC) layer from a two-layer structure into a three-layer structure. Limited by the types of intermetallic compounds (IMCs) generated by Cu-Al, it would be difficult to form a new intermetallic compound (IMC) layer structure with a further increase in annealing temperature, but the thickness of the original intermetallic layer would increase accordingly.

### 3.3. Mechanical Properties and Fracture Analysis

#### 3.3.1. Microhardness

As shown in Figure 10a,b, the microhardness of the weld core zone changes in a gradient due to the presence of copper and aluminum, and the metals on both sides of the copper and aluminum soften to varying degrees with different holding times. The degree of softening increases with increasing holding time, and when the hardness decreases to a certain extent, it will not continue to decrease with increasing holding time. As shown in Figure 10c,d, when comparing the decrease of copper and aluminum microhardness at the same annealing temperature, the decreasing trend of copper side hardness is more significant.

#### 3.3.2. Tensile Strength and Fracture Behavior

The tensile specimen of the lap joint after annealing treatment is shown in Figure 11a,b. It can be seen that it is oxidized to a certain extent following annealing treatment, and there is a shedding oxide scale on the surface. Figure 11c shows the results of the tensile strength test of the lap joint following annealing treatment, when subjected to different annealing temperatures and holding times. The test results show that the tensile strength of the tensile pattern is affected by the annealing process, and that the different annealing processes have different effects on the properties of the joints. When the annealing temperature is 300 °C, there is no obvious change in the strength of the tensile sample with the change of holding time, but when the annealing temperature is 350 °C and 400 °C, the strength of the tensile sample first increases and then decreases with increasing holding time. When the holding time is 2.5, the strength of the sample reaches its maximum. The intermetallic compound (IMC) layer is formed in the dissimilar metal lap joint, and the thickness of the intermetallic compound (IMC) layer is small. After annealing, the intermetallic compound (IMC) layer grows again, and its degree of growth is positively related to the annealing temperature and holding time. Combined with the tensile test results of the joint after annealing, increasing the thickness of the intermetallic compound (IMC) layer in a certain range is beneficial for improving the strength of the joint. However, intermetallic compounds (IMCs) are brittle and hard, and too large an amount of intermetallic compounds (IMCs) will inevitably worsen the performance of the joint and reduce its strength; similarly, too long a holding time leads to a decline in performance.

To understand the effect of the growth of the intermetallic compound (IMC) layer on the fracture behavior of the joint, the fracture surface of the tensile test was further analyzed. The fracture samples along the thermo-mechanical affected zone of the aluminum plate and the fracture samples along the Cu-Al interface were selected to analyze the fracture morphology. Figure 12 shows the fracture morphology of the fracture specimen along the Cu-Al joining interface. There are obvious propagation cracks in the middle of the fracture, which are distributed along the fracture surface and tend to extend to the internal intermetallic compound (IMC) layer. Part of the area was selected for observation with local magnification. Two kinds of fracture characteristics appear on the surface of the break in Figure 12b, with obvious boundaries; the upper surface was relatively flat, and the lower part showed an obvious granular morphology. The EDS test results showed that the selected area was not composed of a single element of copper and aluminum, excluding the possibility of fracture at the substrate. According to the proportion of copper and aluminum, the fracture occurred in the intermetallic compound (IMC) layer. The differences between the copper and aluminum contents at different locations indicate that the fracture does not extend along a single interlayer structure in the intermetallic compound (IMC) layer, but runs through the whole intermetallic compound (IMC) layer.

Figure 13 shows the fracture morphology along the thermomechanically affected zone of the aluminum plate. Table 3 shows the contents of copper and aluminum in different positions. The macroscopic fracture morphology shows that the whole fracture shows obvious regional fracture characteristics. By magnifying and observing different fracture areas, it can be seen that there are intermetallic compounds (IMCs) on the fracture surface near the Cu-Al joining interface, and that the fracture surface is relatively flat. There are a large number of dimples in the middle of the end face of the fracture, which is characteristic of plastic fracture. The end face of the fracture near the surface is fibrous, which is characteristic of plastic fracture. The fracture surface near the Cu-Al joining interface exhibits characteristics of brittle fracture due to the presence of intermetallic compounds (IMCs). The position near the weld surface exhibits characteristics of plastic fracture, as it is located in the thermomechanically affected zone of the aluminum plate, and there is no mixing of copper and aluminum, basically maintaining the original fracture mode of the aluminum plate substrate, that is, plastic fracture behavior. It is not difficult to infer the fracture process of the sample from the fracture characteristics of the fracture surface. The intermetallic compounds (IMCs) generated at the Cu-Al joining interface are brittle and hard. In the process of tension, there is a certain degree of eccentricity on both sides of the plate, which leads to stress concentration at the edge of the joining interface. At this time, the brittle and hard intermetallic compounds (IMCs) cracked first. With the progress of the tensile process, the crack extends to the thermomechanically affected zone of the upper aluminum plate, which has lower strength, eventually leading to the fracture failure of the whole joint. This also explains the reason for the formation of regional fracture characteristics on the fracture surface.

## 4. Conclusions

The effect of post-weld heat treatment on the interfacial structure of Cu-Al and the relationship between mechanical properties were studied. Based on the present analysis, the following conclusions can be drawn:(1)The microstructure characteristics of Cu-Al lap joints after annealing were studied. Annealing treatment can promote the re-growth of intermetallic compounds (IMCs) in the joint. When the annealing temperature is lower and the holding time is short, only the thickness of the intermetallic compound (IMC) layer increases. When the annealing temperature and holding time are increased, the thickness of the intermetallic compound (IMC) layer increases and a new layered structure appears in the middle of the original intermetallic compound (IMC) layer.(2)The effect of the intermetallic compound (IMC) layer on the properties of lap joints was studied. Within a certain range, the performance of the joint can be improved by increasing the holding time of the annealing process and increasing the thickness of the intermetallic compound (IMC) layer.(3)By analyzing the proportion of copper and aluminum elements in different layered structures in intermetallic compounds (IMCs), the intermetallic compound (IMC) composition of different layered structures was determined.(4)The growth process of intermetallic compounds (IMCs) and the reason for the formation of new structures during annealing were revealed. The annealing process is able to increase the activity of atoms and promote the diffusion of atoms. In the process of diffusion, the proportion of copper and aluminum atoms in some regions is different from the initial state. A new layered structure is finally formed under the action of suitable temperature.

## Figures and Tables

**Figure 1 materials-13-04591-f001:**
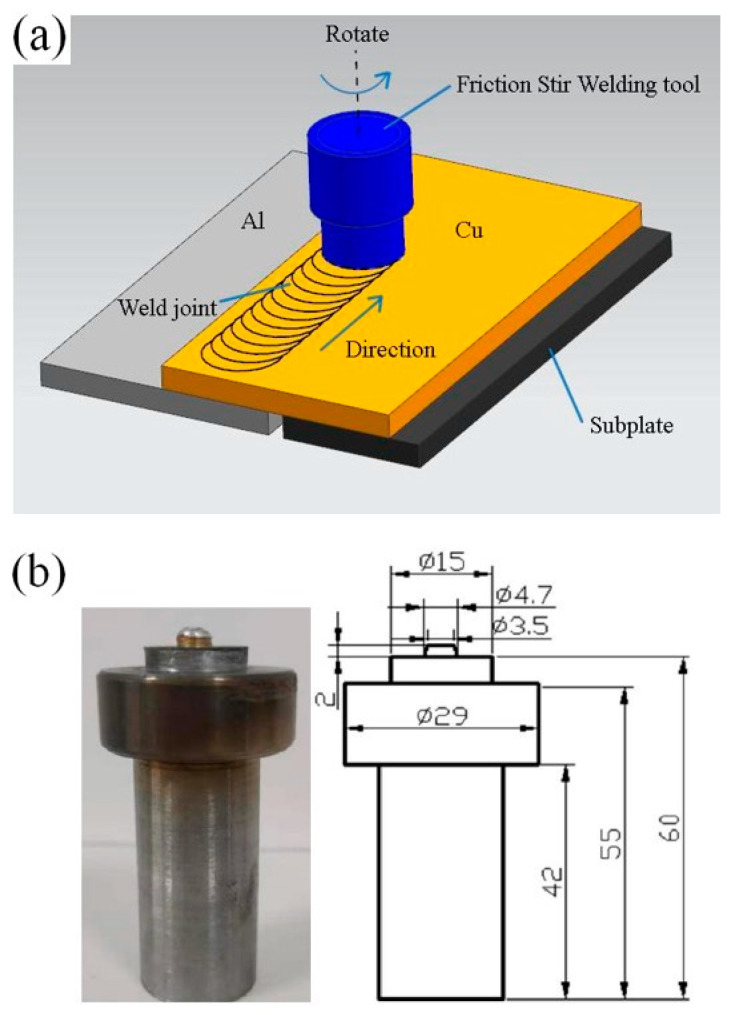
(**a**) Welding process; (**b**) friction stir welding tool (mm).

**Figure 2 materials-13-04591-f002:**
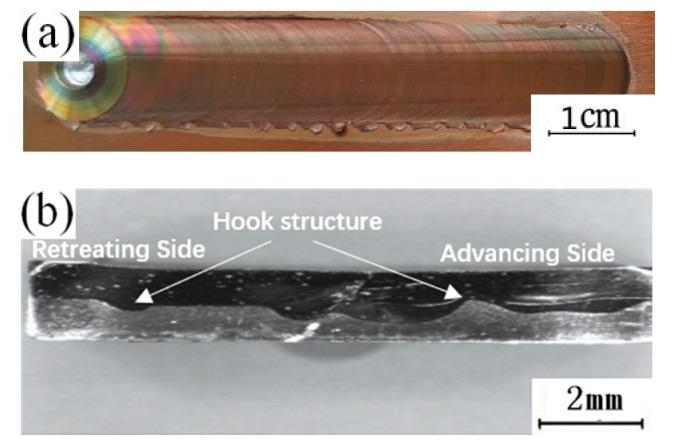
(**a**) Weld surface; (**b**) Weld interior.

**Figure 3 materials-13-04591-f003:**
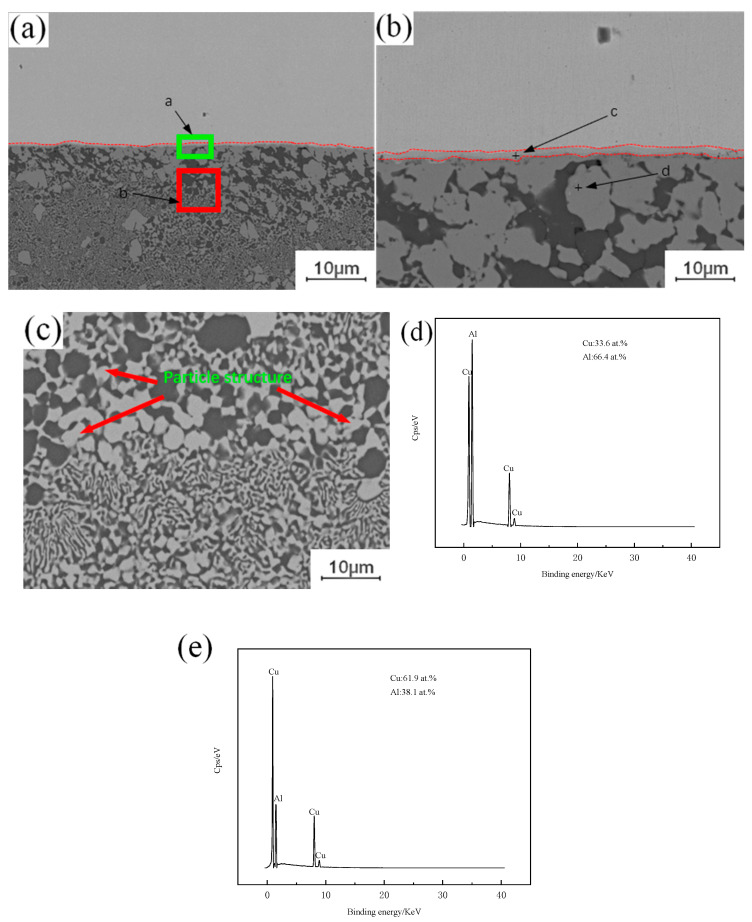
EDS analysis of copper–aluminum lap joint: (**a**) Cu-Al joining interface between copper and aluminum; (**b**) Intermetallic compound (IMC) layer; (**c**) Microstructure of crystal region; (**d**) c point copper and aluminum content; (**e**) d point copper and aluminum content.

**Figure 4 materials-13-04591-f004:**
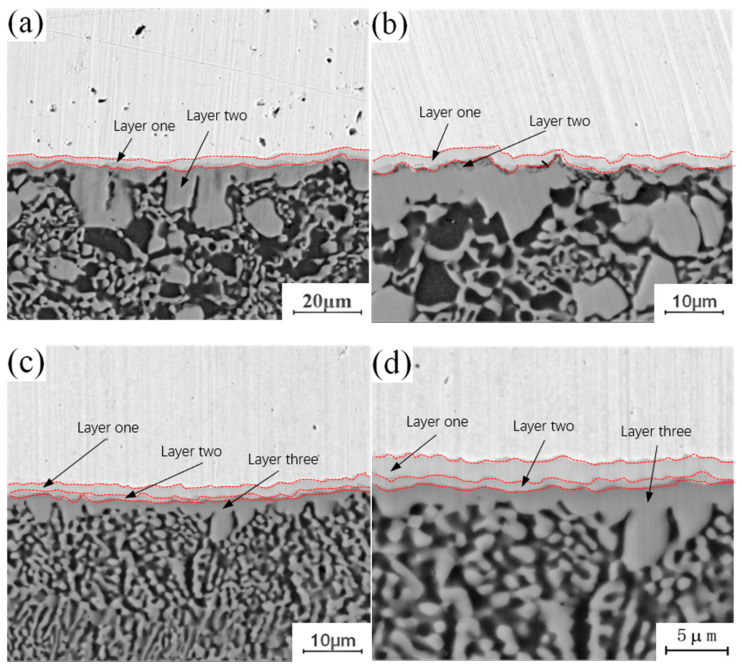
Microstructure of the intermetallic compound (IMC) layer annealed at 300 °C for different holding times: (**a**) t = 0.5 h; (**b**) t = 1.5 h; (**c**) t = 3.5 h; (**d**) t = 4.5 h.

**Figure 5 materials-13-04591-f005:**
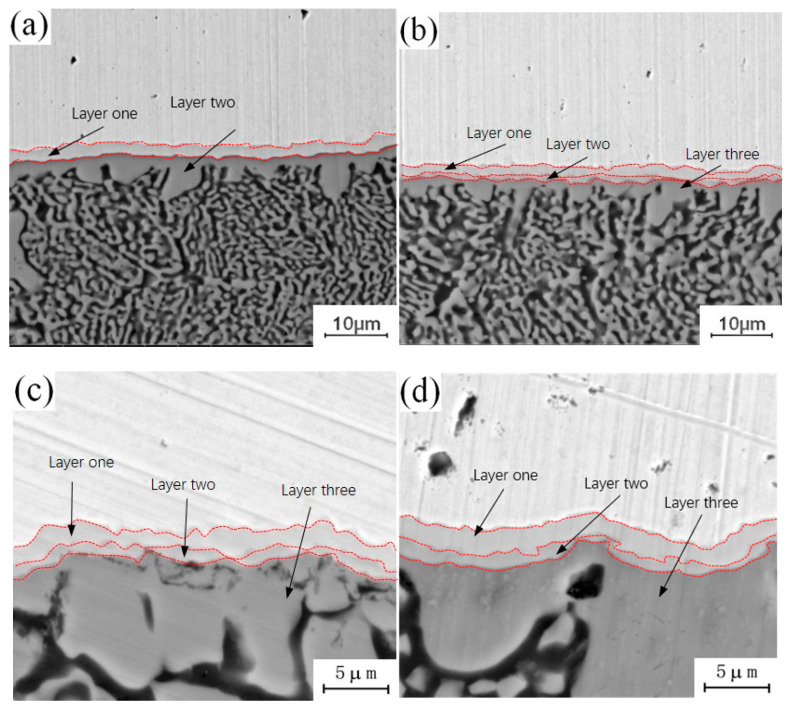
Microstructure of intermetallic compound (IMC) layer annealed at 350 °C for different holding times: (**a**) t = 0.5 h; (**b**) t = 1.5 h; (**c**) t = 3.5 h; (**d**) t = 4.5 h.

**Figure 6 materials-13-04591-f006:**
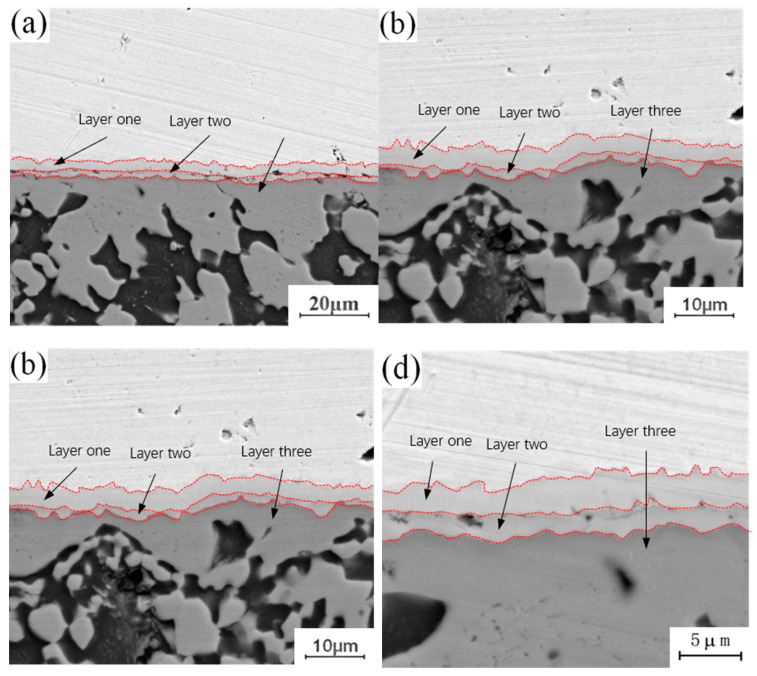
Microstructure of intermetallic compound (IMC) layer annealed at 400 °C for different holding times: (**a**) t = 0.5 h; (**b**) t = 1.5 h; (**b**) t = 3.5 h; (**d**) t = 4.5 h.

**Figure 7 materials-13-04591-f007:**
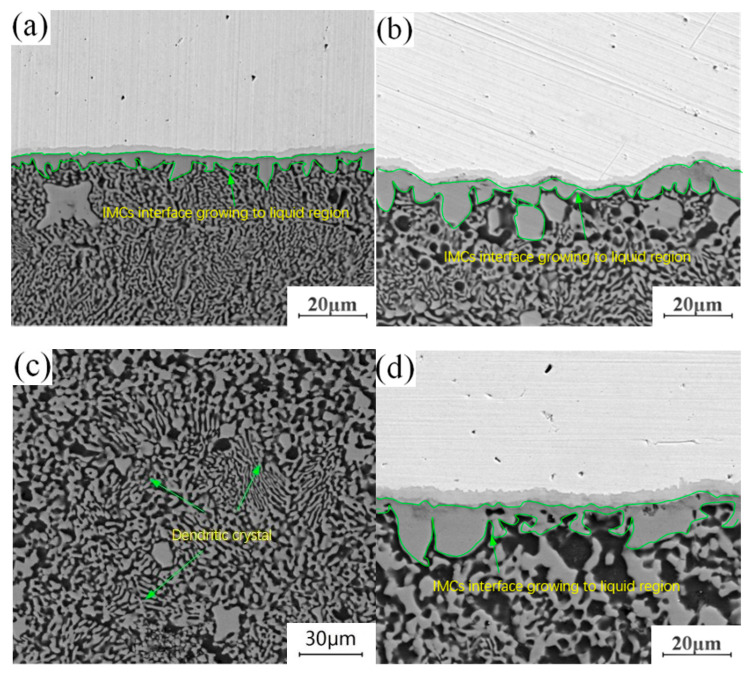
Structural evolution of liquid phase region: (**a**) Unannealed joint; (**b**) Annealing temperature of 300 °C and holding time of 4.5 h; (**c**) Annealing temperature of 350 °C and holding time of 4.5 h; (**d**) Annealing temperature of 400 °C and holding time of 4.5 h.

**Figure 8 materials-13-04591-f008:**
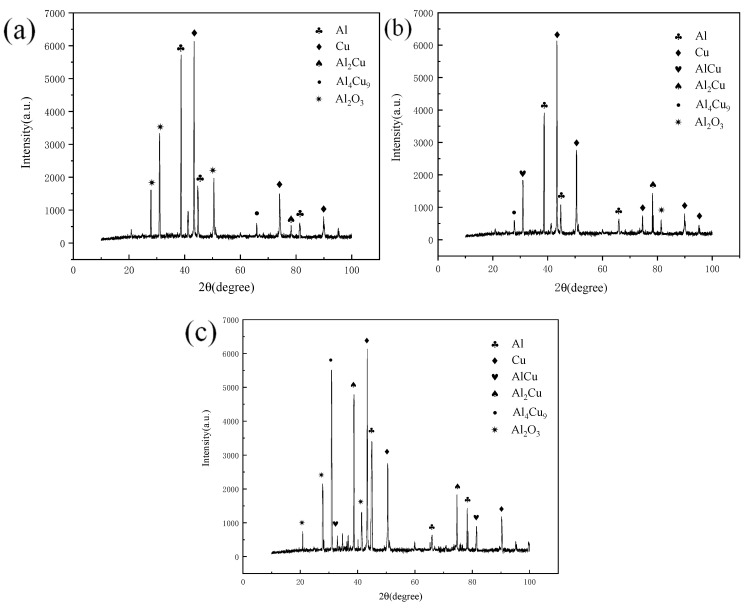
XR diffraction patterns at different annealing temperatures and a holding time of 1.5 h: (**a**) 300 °C; (**b**) 350 °C; (**c**) 400 °C.

**Figure 9 materials-13-04591-f009:**
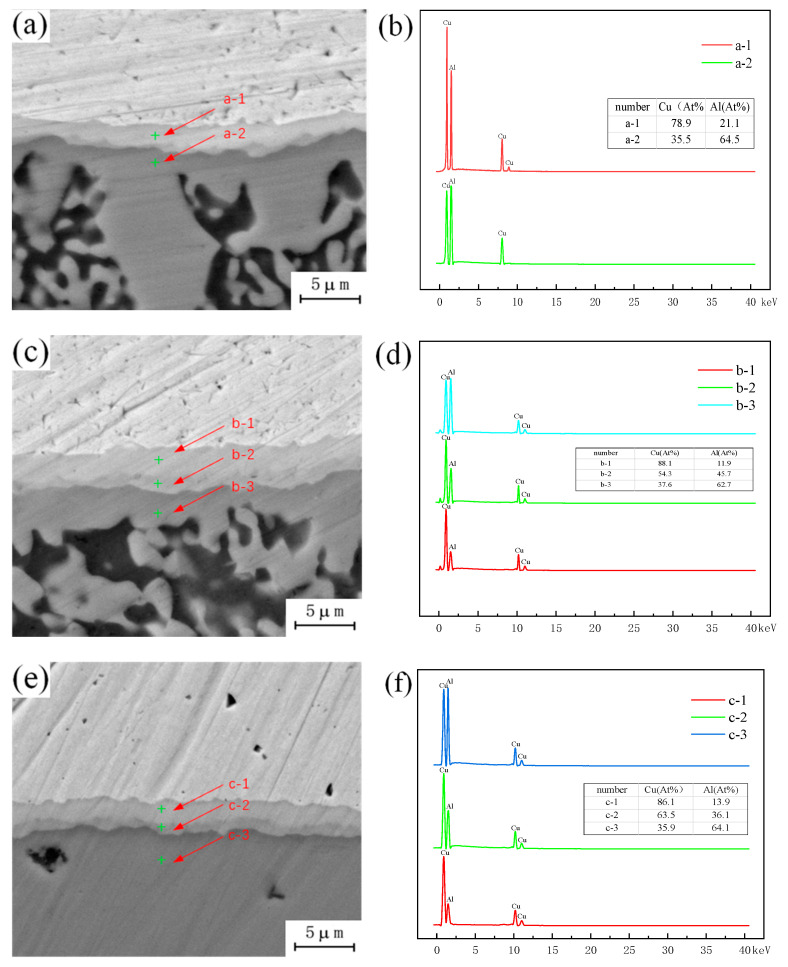
Microstructure of intermetallic compound (IMC) layer under different annealing conditions: (**a**) Annealing temperature 300 °C; (**b**) Content of copper and aluminum in a; (**c**) Annealing temperature 350 °C; (**d**) Content of copper and aluminum in b; (**e**) Annealing temperature 400 °C; (**f**) Content of copper and aluminum in c.

**Figure 10 materials-13-04591-f010:**
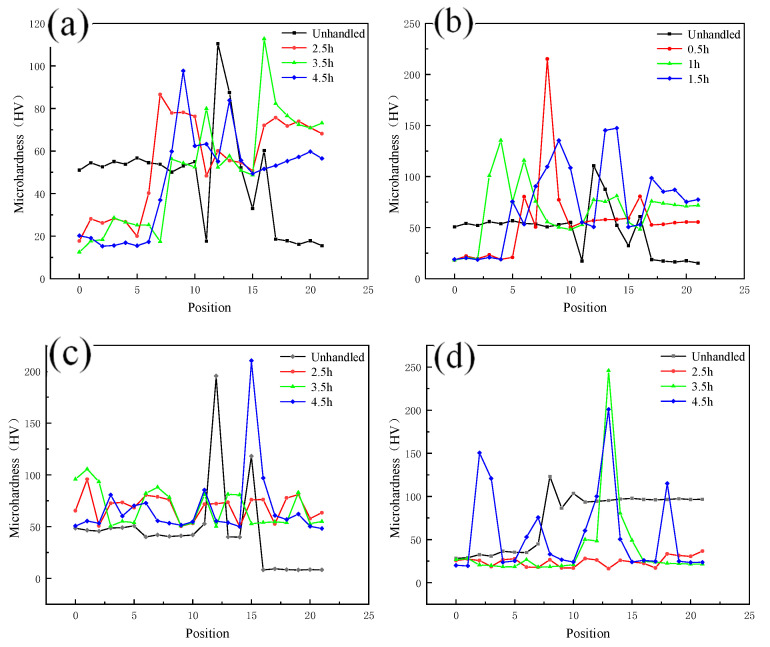
Microhardness of each zone at different annealing temperatures: (**a**) Hardness of nugget zone annealed at 400 °C; (**b**) Hardness of nugget zone annealed at 300 °C; (**c**) Hardness of copper side annealed at 300 °C; (**d**) Hardness of aluminum side annealed at 300 °C.

**Figure 11 materials-13-04591-f011:**
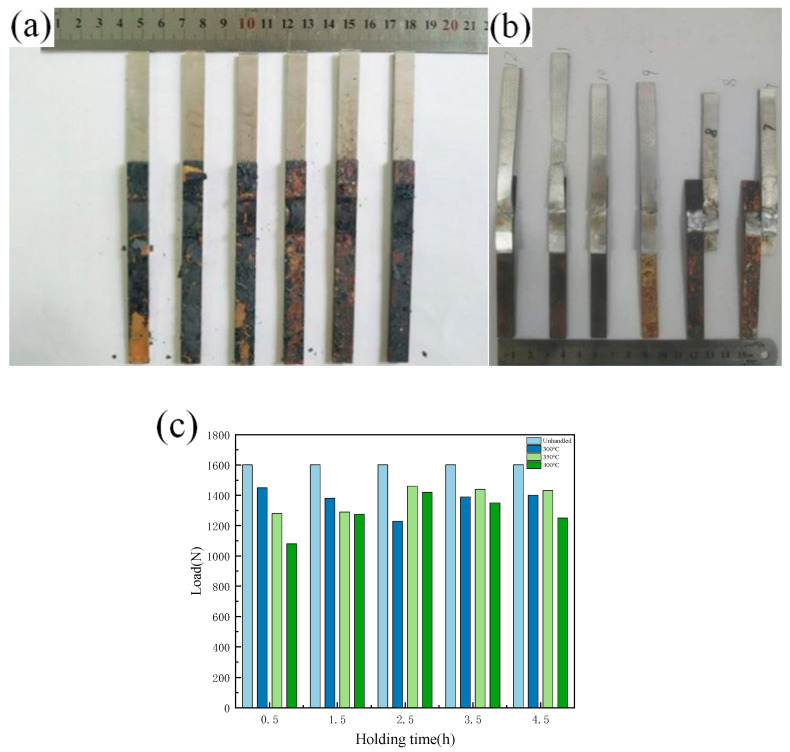
Tensile specimen and tensile strength after annealing treatment: (**a**) Tensile specimen before test; (**b**) Tensile specimen after test; (**c**) Strength of tensile specimen after annealing treatment.

**Figure 12 materials-13-04591-f012:**
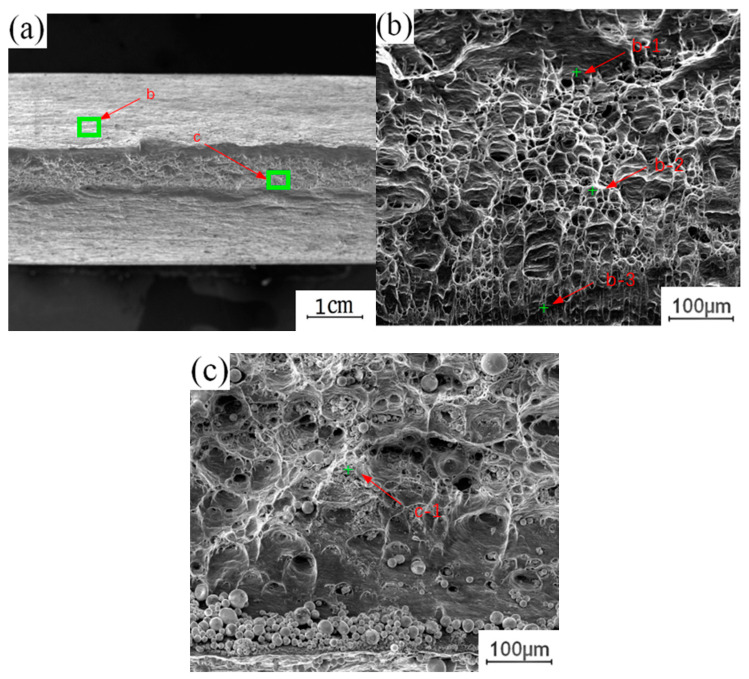
Fracture morphology of the fracture specimen along the Cu-Al joining Interface: (**a**) Macroscopic morphology; (**b**) b morphology; (**c**) c morphology.

**Figure 13 materials-13-04591-f013:**
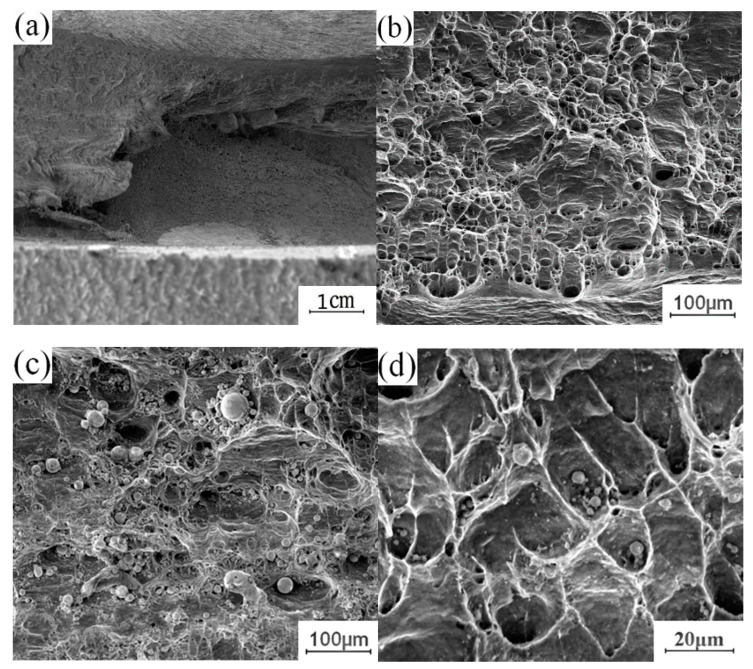
Fracture morphology along the thermomechanical influence zone of aluminum plate: (**a**) Macroscopic morphology; (**b**) Lower morphology; (**c**) Middle morphology; (**d**) Upper morphology.

**Table 1 materials-13-04591-t001:** Chemical constituents of T2 pure copper (wt%).

Element	Cu	Zn	Fe	Ni	Al	P	Si	Mn
Content	99.97	0.003	0.004	0.006	0.009	0.007	0.002	0.001

**Table 2 materials-13-04591-t002:** Chemical constituents of aluminum (wt%).

Element	Cu	Mn	Mg	Zn	V	Ti	Si	Fe	Al
Content	0.05	0.03	0.03	0.05	0.05	0.03	0.25	0.35	99.60

**Table 3 materials-13-04591-t003:** The content of copper and aluminum in different positions.

Element	Cu	Al
b-1	44.7	51.1
b-2	42.6	57.4
b-3	11.5	88.5
c-1	35.1	64.9

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
