# Peer review of "Effect of Post-Weld Annealing on Microstructure and Growth Behavior of Copper/Aluminum Friction Stir Welded Joint"

_materials, 2020, doi:10.3390/ma13204591_

Round 1
Reviewer 1 Report
The paper is of practical interest and the research methodology is appropriate, with relevant investigations method that were used. Conclusions are well supported by the experimental results, which were correctly interpreted.
I suggest some formal modifications to be performed.
- Minor errors to be corrected ; for example, Table 2, Figure 6; a thorough revision would be useful.
- Symbolization of layers is somehow confusing: when Layer three appears, Layer 2 develops to layer 3? Try another method
- Double check of language style and spelling
- Abbreviation should be explained in the text (ex: IMC – intermetallic compound; FSW – friction stir welding etc.)
Author Response
Dear Editors and Reviewers:
Thank you for your letter and for the reviewers’comments concerning our manuscript entitled “Effect of post-weld annealing on microstructure and growth behavior of copper / aluminum friction stir welded joint”.Those comments are all valuable and very helpful for revising and improving our paper, as well as the important guiding significance to our researches. All the revised parts have been revised in the manuscript. The following is a point-to-point response to your comments.
Responds to the reviewer’s comments:
Reviewer #1:
- Response to comment:( Minor errors to be corrected ; for example, Table 2, Figure 6; a thorough revision would be useful)
Respones: Table 2 has been modified. Figure 6 is the question of the ruler?
- Response to comment:( Symbolization of layers is somehow confusing: when Layer three appears, Layer 2 develops to layer 3? Try another method)
Respones: The second layer did not become the third layer, but the third layer grew between the first layer and the second layer. At the fixed annealing temperature, with the increase of holding time, the original structure thickness of the original IMCS layer will increase, and a new layered structure appears in the middle of the original structure, and its composition is obviously different from the original structure.At a short temperature and time, only the original IMCS layer structure grows, but no new structure appears, because at a fixed annealing temperature and a short holding time, the diffusion number and range of copper and aluminum atoms are limited, which can not reach the conditions for the formation of new layered structures. with the increase of holding time, the diffusion number and diffusion range of copper and aluminum atoms also increase. At this time, the middle position of the original structure satisfies the material and thermodynamic conditions for the formation of a new layered structure, thus promoting the formation of a new layered structure.
- Response to comment:( Double check of language style and spelling)
Respones: We examined the language and spelling carefully
- Response to comment:( Abbreviation should be explained in the text (ex: IMC – intermetallic compound; FSW – friction stir welding etc)
Respones: We have explained the abbreviations in the text.
Special thanks to you for your good.
Reviewer 2 Report
An interesting paper. Presented results and relation between annealing temperature and holding time are expectable. The inter layer is good presented what is most important in this case. Explanation of IMCs layer formation must be corrected.
I suggest to add bigger or additional figures with the macrostructure of the joint in cross-section perpendicular to welding direction.
For welded joints I prefer the word: joint or welded joint in spite of connection. Using connection is also acceptable.
Line 102 – heat preservation time – in my opinion it is “holding time” or “annealing time”.
Line 113 - Table 2. Chemical constituents of T2 pure copper(wt%) - it is chemical composion of aluminum
Fig. 2b – what it is “hock structure”? may be “hock-shape structure” or irregular fusion line?
What it is a kind of defect: tunnel inside? Can you show in which standard such defect is classified.
Line 136 and further – you are using: the copper-aluminum bonding interface. Bonding is he specific process of welding where it is needed an addition of filler metal with melting point lower then base metals. In your case, I suppose, that there is the mixing zone or fusion zone. And it is an effect of melting and crystallization of heat input (what is presented in lines 148-155).
Line 155/158 – should be CuAl2
Line 260-262 – how oxides can be formed during annealing inside the material? In my opinion, the surface of metals before FSW was polluted – in this case oxides are before annealing.
Author Response
Dear Editors and Reviewers:
Thank you for your letter and for the reviewers’comments concerning our manuscript entitled “Effect of post-weld annealing on microstructure and growth behavior of copper / aluminum friction stir welded joint”.Those comments are all valuable and very helpful for revising and improving our paper, as well as the important guiding significance to our researches. All the revised parts have been revised in the manuscript. The following is a point-to-point response to your comments.
Responds to the reviewer’s comments:
Reviewer #2:
- Response to comment:( Explanation of IMCs layer formation must be corrected.)
Respones: According to the phase diagram of aluminum and copper, the intermetallic compound of aluminum and copper can be formed at about 200 ℃. Therefore, intermetallic compounds are bound to be produced near the interface in the welding process accompanied by intense material mixing or suitable heat treatment conditions.
- Response to comment:( I suggest to add bigger or additional figures with the macrostructure of the joint in cross-section perpendicular to welding directi on)
Respones: We have tried to add this kind of graphics before, but the addition is a bit complicated and unattractive.
- Response to comment:( For welded joints I prefer the word: joint or welded joint in spite of connection. Using connection is also acceptable)
Respones: We have considered using “connection” before, but many references use “welded joints”, so we also use “welded joints”.
- Response to comment:( Line 102 – heat preservation time – in my opinion it is “holding time” or “annealing time”)
Respones: We have changed it to “annealing time”.
- Response to comment:( Line 113 - Table 2. Chemical constituents of T2 pure copper(wt%) - it is chemical composion of aluminum)
Respones: We have changed it to “Chemical constituents of aluminum”.
- Response to comment:( Fig. 2b – what it is “hock structure”? may be “hock-shape structure” or irregular fusion line?)
Respones: We have changed it to “hock-shape structure”.
- Response to comment:( What it is a kind of defect: tunnel inside? Can you show in which standard such defect is classified)
Respones: The tunnel defect has a significant effect on the quality of the joint, which not only reduces the strength of the joint, but also deteriorates the bearing capacity of the structure, but also affects the sealing performance of the product. Its structural and morphological features are as follows: continuous linear holes along the welding direction inside the weld. Figure 3 is the x-ray picture of the tunnel defect, and figure 4 is the metallographic picture of the cross section of the tunnel defect in the weld.

Reviewer 3 Report
Authors reported their research work on “Effect of post-weld annealing on microstructure and growth behavior of copper / aluminum friction stir welded joint”. The work is well organized and well explained. Few minor revisions needed: 1. The red arrow mark and green boxes are not prominent. Authors should increase the width of the line and outline of the box, so it will be easier to understand. 2. In Reference section, authors put reference no. 2 times. Please check and modify. 3. Authors may include few more references available on the Al-Cu FSW work in the introduction section. As an example: https://link.springer.com/article/10.1007/s00170-016-8820-0.Author Response
Dear Editors and Reviewers:
Thank you for your letter and for the reviewers’comments concerning our manuscript entitled “Effect of post-weld annealing on microstructure and growth behavior of copper / aluminum friction stir welded joint”.Those comments are all valuable and very helpful for revising and improving our paper, as well as the important guiding significance to our researches. All the revised parts have been revised in the manuscript. The following is a point-to-point response to your comments.
Responds to the reviewer’s comments:
Reviewer #3:
- Response to comment:( The red arrow mark and green boxes are not prominent. Authors should increase the width of the line and outline of the box, so it will be easier to understand.)
Respones: We have increased the width of the line and outline of the box.
- Response to comment:( In Reference section, authors put reference no. 2 times. Please check and modify.)
Respones: We have modified it to “Aluminum and copper are widely used in aerospace, electronics, chemical and other fields because of their good thermal conductivity and electrical conductivity. [1-4].”
- Response to comment:( Authors may include few more references available on the Al-Cu FSW work in the introduction section. As an example: https://link.springer.com/article/10.1007/s00170-016-8820-0.)
Respones: Al-Cu FSW work is already included in the references, and too much addition may make the introduction too cumbersome.
Special thanks to you for your good.
